Geometrical determinants of cerebral artery fenestration for cerebral infarction

Mei Yuqian mei.yuqian@nsmc.edu.cn 1
Chen Xiaoqin 2
Zhang Yao 1
Wang Yanling 1
Wu Bo 3
Hu Mingcheng 4
Bao Quan 4
1 School of Medical Imaging, North Sichuan Medical College , Nanchong , Sichuan , China
2 Department of Radiology, West China Hospital, Sichuan University , ChengDu , Sichuan , China
3 North Sichuan Medical College, Academic Affairs Office , Nanchong , Sichuan , China
4 Department of Magnetic Resonance Imaging, Hongqi Hospital, Mudanjiang Medical University , Mudanjiang , Heilongjiang , China
Marunaka Yoshinori
Electronic publication date: 2025 Jan 21
Publication date: 2025
Volume: 13
Electronic Location ID: e18774
Received 2024 Sep 10; Accepted 2024 Dec 6
Copyright: ©2025 Mei et al.
Copyright year: 2025
Copyright holder: Mei et al.
License: This is an open access article distributed under the terms of the Creative Commons Attribution License, which permits unrestricted use, distribution, reproduction and adaptation in any medium and for any purpose provided that it is properly attributed. For attribution, the original author(s), title, publication source (PeerJ) and either DOI or URL of the article must be cited.
License URL: https://creativecommons.org/licenses/by/4.0/

Keywords: Atherosclerotic plaque, Cerebral artery fenestration, Cerebral infarction, Hemodynamics, Morphological parameters, Regression predictive model

Funding: Young Scientists Fund of the Natural Science Foundation of Sichuan Province of China 2024NSFSC1705 Doctoral Scientific Research Foundation of North Sichuan Medical College CBY21-QD03 Fostering Distinguished Young Scholars CBY22-JQ02 “HongQi Research Fund” science and technology project 2019HQ-10 This work was supported by the Young Scientists Fund of the Natural Science Foundation of Sichuan Province of China (2024NSFSC1705), the Doctoral Scientific Research Foundation of North Sichuan Medical College (CBY21-QD03), the Fund for Fostering Distinguished Young Scholars (CBY22-JQ02) and the “HongQi Research Fund” science and technology project (2019HQ-10). The funders had no role in study design, data collection and analysis, decision to publish, or preparation of the manuscript.

==============================
Purpose

Few data are available on the causality of cerebral artery fenestration (CAF) triggering cerebral infarction (CI) and this study aims to identify representative morphological features that can indicate risks.

Methods

A cohort comprising 89 patients diagnosed with CAF were enrolled from a total of 9,986 cranial MR angiographies. These patients were categorized into Infarction Group (n = 55) and Control Group (n = 34) according to infarction events. These two groups are divided into two subgroups depending on fenestration location (basilar artery or other cerebravascular location), respectively, i.e., BA Infarction Group (n = 37), BA Control Group (n = 23), Non_BA Infarction Group (n = 18), Non_BA Control Group (n = 11). This study firstly defined 12 indices to quantify the morphological characteristics of fenestration per se and its connecting arteries. The data were evaluated using either the independent sample t-test or the Mann–Whitney U test. Conducting univariate and multivariate logistic regression analyses to ascertain potential independent predictors of CI.

Results

The initiation angle φ1 and confluence angle φ2 at the fenestration in the Infarction Group are both smaller compared to the Control Group, but only the Infarction Group and BA Infarction Group have significant difference (p < 0.05). The maximum left fenestration axis (fAL) and the left tortuosity index (TIL) were greater in the Infarction Group for CAFs than those in the Control Group (p < 0.05). In contrast, the maximum right fenestration axis (fAR) and the right tortuosity index (TIR) were smaller than those in Control Group (p < 0.05). The logistic regression analysis revealed that φ2 (AUC = 0.68, p = 0.02), fAL (AUC = 0.72, p < 0.01), and fAR (AUC = 0.70, p < 0.01) serve as independent risk factors influencing the occurrence of CI. The regression predictive model achieved an AUC of 0.83, enabling accurate classification of 77.5% of cases, indicating a robust predictive performance of the model.

Conclusion

Morphological results demonstrated a left-leaning type of fenestration with more narrow fenestration terminals indicating a higher risk of CI occurrence. Furthermore, the regression predictive model established in this study demonstrates a good predictive performance, enabling early prediction of CI occurrence in fenestrated patients and facilitating early diagnosis of CI.

Introduction

Cerebral artery fenestration (CAF) is a congenital vascular anomaly resulting from the failed fusion of primitive embryonic vessels. In the microscopic view, the local replication of endothelial cells and medial cleavage leads to the formation of a single lumen by splitting the arterial lumen and subsequent distal reconnection (Gailloud et al., 2002; Kathuria et al., 2011). CAF can occur in multiple vessels, with a predilection for the anterior cerebral artery (ACA) and the basilar artery (BA) (Gold & Crawford, 2013). The prevalence of the latter ranges from 0.3% to 0.6% in conventional angiography and from 1% to 2.07% in magnetic resonance angiography (Sanders, Sorek & Mehta, 1993; Uchino et al., 2001). CAFs are often overlooked in clinical practice due to their small sizes, i.e., 1 mm–5 mm typically, representing a frequently neglected cerebrovascular anomaly (Sogawa et al., 2013). While asymptomatic CAF are incidentally discovered during other medical evaluations more oftentimes. Therefore, early detection, and timely and effective management, are hard to achieve in the initial stages of development, especially for pathological CAF, which might induce subsequent severe outcomes (Wang & Ge, 2021).

Although CAFs are relatively rare, they have been reported in association with cerebral infarctions (CI), aneurysms, arteriovenous malformations, and subarachnoid haemorrhages (SAHs) (Kloska et al., 2006; Palazzo et al., 2014; Trivelato et al., 2016). In a study of 20 patients with unexplained SAH, fenestrations were detected near thrombotic locations. This suggests that fenestrations may be a weakness in arterial structure and have an adverse impact on disease development (Hudák et al., 2013). However, studies regarding the causality between fenestrations and cerebral infarction are limited. Cerebral blood vessels play a crucial role in maintaining normal physiological circulation in the brain. Irregularities in the structure of cerebral blood vessels, like the occurrence of fenestrations, can disrupt the blood supply to brain tissue, and hinder the timely establishment of collateral circulation (Patel et al., 2014). More seriously, it may further lead to abnormal hemodynamic changes, ultimately resulting in cerebral tissue hypoxia and progressing to cerebral infarction (Hirai et al., 2021). This can lead to clinical syndromes characterized by neurological deficits, significantly affecting the life quality of patients (Malek, Alper & Izumo, 1999). The exact mechanisms underlying fenestration-related infarctions remain unclear but are hypothesized to be related to alterations in the hemodynamic environment due to the dual-channel blood flow pathway created by fenestrations.

CAFs can change the geometric characteristics of arteries, which inevitably affect the pattern blood flows through them, such as the formation of high-viscosity blood and blood stasis. Pathologically, the disturbed flow can affect the biological functions of endothelial cells lining the inner arterial walls, promoting thrombotic events and contributing to CI (Sui et al., 2008; Chen et al., 2021). Therefore, analyzing the morphological parameters of CAFs is essential for understanding their role in the formation of CI.

In this study, we retrospectively included 89 cases of CAF, extracted morphological parameters from reconstructed models, and analyzed the geometric characteristics of along-fenestrated regions. The objective of this study is to discriminate unfavorable morphological characteristics of fenestrations which are potential markers for the formation of CI.

Materials and Methods

Data acquisition

In this comprehensive investigation, from April 2020 to November 2022, a total of 9,986 cranial MR angiographies were performed at our hospital. We recruited a cohort comprising 89 patients diagnosed with CAF who underwent magnetic resonance angiography (MRA) scans (3D TOF sequence without contrast). The research protocol was reviewed and approved by the Ethics Committee of Mudanjiang Medical University (Approval No. 202054). Written informed consent was obtained from all study participants or their legal guardians. Inclusion criteria encompass the following conditions (Zhu et al., 2011): (1) unequivocal diagnosis of CAF via cranial MRA; (2) the cerebral infarction cohort, defined by neurological findings on cranial MRI consistent with cerebral infarction; (3) the cerebral infarction area supplied by fenestrated vessels; (4) the control group, devoid of cerebral infarction, ascertained through cranial MRI assessment; and (5) the availability of high-quality MRA reconstruction models permitting subsequent hemodynamic analysis. Exclusion criteria were also implemented, eliminating patients (Ahmetgjekaj et al., 2014): (1) afflicted with other vascular anomalies or intracranial neoplasms potentially influencing arterial blood flow; and (2) concurrently suffering from severe cardiac, pulmonary, hepatic, or renal dysfunction. All examination data were uniformly stored in the internationally recognized DICOM format on optical discs. After exclusion of above reasons, we also eliminated 11 subjects due to incomplete vascular reconstructions with poor reconstruction quality (n = 3), incomplete central lines (n = 3), and significant image artefacts (n = 5), resulting in subpar image quality. The whole exclusion process and illustration of representative exclusion cases were listed in Fig. 1. These datasets were categorized into six groups referring to fenestration location and infarction event, denominated as follows (as shown in Fig. 1):

Figure 1 Schematic representation of study population grouping and selection criteria.

1. BA Infarction Group (n = 37): patients encountered infarction whose fenestration was located at the BA;

2. BA Control Group (n = 23): fenestration located at the BA without infarction event;

3. Non_BA Infarction Group (n = 18): patients encountered infarction whose fenestration was located at the ACA, vertebral artery (VA), anterior communicating artery (ACOA) and middle cerebral artery (MCA);

4. Non_BA Control Group (n = 11): the location of fenestration was the same as Non_BA Infarction group but without infarction;

5. Infarction Group (n = 55): all infarction patients (BA Infarction group + Non_BA Infarction group);

6. Control Group (n = 34): all control patients (BA Control group + Non_BA Control group).

The cutting-edge imaging technology was harnessed for this purpose, employing the Philips Intera 3.0 T superconducting MR scanner paired with an 8-channel head coil. Standard cranial magnetic resonance imaging (MRI) and MRA data were judiciously collected. The parameters for the MRA scans were as follows: a repetition time (TR) of 32 ms, an echo time (TE) of 4 ms, a slice thickness of 5 mm, an interslice spacing of 1 mm, a matrix size of 256 × 256, and a field of view (FOV) measuring 240 mm ×240 mm.

Model reconstruction

Two radiologists with 6 and 8 years of experience in head MRI and MRA, respectively, reviewed the examination images (to check for the matching of fenestrated vessels and cerebral infarction areas, as well as image quality), followed by manual segmentation of the images by two other radiologists with 9 and 10 years of experience in head MRI and MRA, ensuring that the segmenting radiologists had no prior subjective impressions of the images (Kathuria et al., 2011; Gold & Crawford, 2013). All aforementioned radiologists were blinded to clinical information and the subsequent vascular parameter information of the patients. The final morphological parameter extraction was conducted by two additional radiologists, one of whom is a co-author, X.Q.C., with 5 years of experience as an imaging physician. They were unaware of the previous imaging results of the patients. All constructed fenestration models were allocated into three shape types and the detail information was presented in Table 1 and Fig. 2.

Table 1 Shape and location of different fenestrations.

Shape of fenestrations	BA	ACA	VA	ACOA	MCA	Total	
Slit-liked	53	11	2	2	1	69	
Convex-lens-like	5	6	5	0	0	16	
Large fenestration (symmetric)	2	0	2	0	0	4	
Total	60 (67.4%)	17 (19.1%)	9 (10.1%)	2 (2.3%)	1 (1.1%)	89	
Notes.

BA basilar artery

ACA anterior cerebral artery

VA vertebral artery

ACOA anterior communicating artery

MCA middle cerebral artery

Figure 2 Imaging results showing fenestrations in five different locations.

(A–E) Fenestrations located in the (A) basilar artery (BA), (B) anterior cerebral artery (ACA), (C) vertebral artery (VA), (D) anterior communicating artery (ACoA), and (E) middle cerebral artery (MCA).

Morphological parameter measurement

The centerline of the CAF model was generated and imported into MATLAB (MathWorks, Natick, MA, USA) for the calculation of morphological parameters. Specific definitions are illustrated in Fig. 3 (Zhang et al., 2018; Zhang et al., 2019a; Ngo, Kwak & Chung, 2020; Zhang et al., 2019b):

Figure 3 (A–D) Definition of morphological parameters.

(1) Initiation angle φ1/confluence angle φ2: angle formed by two tangent vectors along the bifurcation centerlines at the start and end points of the fenestration respectively.

(2) Fenestration linear distance Df: The linear distance from the initiation point to the termination point of the fenestration.

(3) Fenestration involved distance Dfi: The linear distance from the initiation point to the ending point of the artery segment that includes the fenestration, e.g., the A1 segment of ACA.

(4) Maximum left/right fenestration axis fAL/fAR: The maximum distance from points along the left/right fenestration centerline to the fenestration linear line Df.

(5) Total fenestration axis fAT: the sum of left and right maximum fenestration axis (fAL + fAR).

(6) Path lengths LPL/LPR: The actual vascular lengths from the initiation to the termination points along the vascular centerline on the left/right sides of the fenestration.

(7) Left/right tortuosity index: TIL/R = (LPL/R/Df -1) ×100%.

(8) Maximum fenestration vessel height ratio: RH = Df/Dfi ×100%.

(9) Fenestration vessel length-to-diameter ratio: RL/D = Df/fAT ×100%.

(10) Left dominance of fenestration arterial vessels: fAL > fAR and TIL > TIR

(11) Right dominance of fenestration arterial vessels: fAR > fAL and TIR > TIL

Statistical analysis

SPSS (Version 26.0, IBM Corp., Armonk, NY, USA) was used to analyze the data. The categorical variables are presented in terms of frequency (percentage), and continuous variables undergo tests for normality and homogeneity of variance, expressed as either mean ± standard deviation or median (quartiles). The data were evaluated using either the independent sample t-test or the Mann–Whitney U test. Univariate and multivariate logistic regression analyses were performed to identify potential independent predictors of CI. Significant variables from univariate analysis are included in the multivariate logistic regression model. Receiver operating characteristic curve (ROC) analysis was conducted to assess the model’s performance, calculate the area under the curve (AUC), sensitivity, and specificity. Statistical significance is determined using a two-sided test, with p < 0.05 indicating significance.

Results

Study population and descriptive characteristics

A retrospective analysis was conducted on 89 patients diagnosed with CAF. The average age of the Infarction Group (n = 55) was 60.18 ± 9.96 years, comprising 35 male and 20 female patients, while the Control Group (n = 34) had an average age of 57.24 ± 11.30 years, including 18 male and 16 female patients. Among the participants, 60 patients belonged to the BA subgroup, and 29 to the non-BA subgroup. In the BAInf subgroup (n = 37), the average age was 59.84 ± 8.78 years, with 23 male and 14 female patients. The BACon subgroup (n = 23) had an average age of 57.13 ± 10.98 years, including 13 male and 10 female patients. In the Non_BAInf subgroup (n = 18), the average age was 60.89 ± 12.29 years, comprising 12 male and six female patients, while the Non-BA Con subgroup (n = 11) had an average age of 57.45 ± 12.49 years, with five male and six female patients. Baseline characteristics are summarized in Table 2.

Table 2 Baseline characteristics of included subjects.

Variables	BA Infarction Group (n = 37)	BA Control Group (n = 23)	p	Non_BA Infarction Group (n = 18)	Non_BA Control Group (n = 11)	p	Infarction Group (n = 55)	Control Group (n = 34)	p	
Age (years)	59.84 ± 8.78	57.13 ± 10.98	0.30	60.89 ± 12.29	57.45 ± 12.49	0.47	60.18 ± 9.96	57.24 ± 11.30	0.20	
Male sex, n (%)	23 (62%)	13 (57%)	0.67	12 (67%)	5 (45%)	0.44	35 (64%)	18 (53%)	0.32	
Current smoking, n (%)	22 (59%)	12 (52%)	0.58	10 (56%)	6 (55%)	1.00	32 (58%)	18 (53%)	0.63	
Diabetes mellitus, n (%)	24 (65%)	11 (48%)	0.19	13 (72%)	7 (64%)	0.69	37 (67%)	18 (53%)	0.18	
Hypertension, n (%)	19 (51%)	11 (48%)	0.79	12 (67%)	6 (55%)	0.70	31 (56%)	17 (50%)	0.56	
Dyslipidemia, n (%)	24 (65%)	14 (61%)	0.76	11 (61%)	5 (45%)	0.47	35 (64%)	19 (56%)	0.47	

Subgroup comparisons regarding CI and vascular conditions

(1) The case numbers regarding the left dominance of fenestration arterial vessels were as follows: In the Infarction Group, there were 37 cases in total, in the BA Infarction Group there were 23 cases, and in the Non_BA Infarction Group there were 14 cases. In the Control Group, there were eight cases in total, in the BA Control Group there were seven cases, and in the Non_BA Control Group there was one case. The specific numbers regarding right dominance of arterial vessel fenestrations have been included in Table 3. The results showed that the number of cases with left dominance of fenestration arterial vessels in the Infarction Group and its subgroups exceeds that of the Control Group.

Table 3 Distribution of atherosclerosis.

Distribution of atherosclerosis	BA Control Group (n = 23)	Non_BA Control Group (n = 11)	BA Infarction Group (n = 37)	Non_BA Infarction Group (n = 18)	Control Group (n = 34)	Infarction Group (n = 55)	
Left dominance of arterial vessels	7	1	23	14	8	37	
Right dominance of arterial vessels	16	10	14	4	26	18	
Atherosclerosis	11	4	34	14	15	48	
No atherosclerosis	12	7	3	4	19	7	

(2) Among the BA Infarction Group, two cases were mixed-type infarctions, and 35 cases were non-hemorrhagic. All 18 cases in the Non_BA Infarction Group were non-hemorrhagic. In these 55 cases of CI patients, all brain tissue regions supplied by fenestrated vessels exhibited CI. Specifically, strokes occurred in the vertebrobasilar artery system (VAS) region at a rate of 76.36% (42/55). Typical radiographic features of CI patients are presented in Fig. 4.

Figure 4 Imaging results of four typical cases showing stroke lesions located in the arterial fenestration area (highlighted by yellow arrows indicating lesions and fenestrations).

Cases 1–4 all involve brainstem infarctions, and MRA reveals fenestrations located at the vertebral-basilar junction or proximal segment of the basilar artery (BA). A–B, C–D, E–F, G–H correspond to the same patient.

(3) Regarding atherosclerosis, the results showed that the number of cases with atherosclerosis in the BA Infarction Group (34/60) was higher than in the BA Control Group (11/60). Similarly, the number of cases with atherosclerosis in the Non_BA Infarction Group (14/29) was higher than in the Non_BA Control Group (4/29). Furthermore, the number of cases with atherosclerosis in the Infarction Group (48/89) was higher than in the Control Group (15/89), as detailed in Table 3.

Comparison of morphological parameters between different groups

The statistical analysis revealed that fAL and TIL in BA Infarction Group were both greater than those in the Control Group (p < 0.05), while φ1, φ2, fAR, and TIR were all smaller than that in BA Control Group (p < 0.05).

Similar to the BA Infarction Group, the Non_BA Infarction Group also demonstrated that fAL and TIL showed greater values compared to Non_BA Control Group (p < 0.05), whereas fAR, and TIR were smaller in the Non_BA Infarction Group (p < 0.05). However, there is no significant difference about φ1 and φ2.

In the Infarction Group, it was observed that the magnitude of fAL and TIL were both higher than that in Control Group (p < 0.05), while values of φ1, φ2, fAR, and TIR were all lower in comparison with Control Group (p < 0.05), possibly due to the smaller number of Non-BA cases in this study, resulting in a bias towards the BA group.

Detailed parameter values for each group are presented in Table 4.

Table 4 Analysis of morphological parameters.

Morphological parameters	BA infarction group (n = 37)	BA control group (n = 23)	p	Non_BA infarction group (n = 18)	Non_BA control group (n = 11)	p	Infarction group (n = 55)	Control group (n = 34)	p	
φ1 (∘)	84.06 ± 15.66	95.81 ± 20.63	0.02*	86.39 (81.29, 98.04)	91.06 (89.08, 98.22)	0.35	85.80 (78.36, 93.73)	96.10 (89.54, 102.74)	0.00*	
φ2 (∘)	83.64 (77.35, 97.25)	98.05 (93.67, 105.00)	0.01*	90.13 (83.82, 94.97)	92.68 (86.77, 96.02)	0.50	87.1 (80.42, 96.72)	96.3 (89.50, 99.60)	0.01*	
Df (mm)	5.59 (5.14, 6.41)	5.76 (4.95, 8.57)	0.54	8.20 (5.35, 9.71)	8.22 (5.45, 9.02)	0.75	5.93 (5.21, 8.61)	7.23 (5.18, 8.66)	0.76	
Dfi (mm)	27.40 (24.06, 29.51)	27.44 (22.80, 29.51)	0.84	17.41 (16.13, 29.03)	17.68 (12.55, 31.75)	0.79	26.4 (19.77, 29.30)	26.9 (19.81, 30.95)	0.96	
fAL (mm)	1.95 (1.34, 2.68)	1.43 (0.66, 1.96)	0.01*	2.36 (2.14, 3.54)	1.47 (0.92, 2.19)	0.01*	2.10 (1.38, 2.82)	1.45 (0.77, 2.04)	0.00*	
fAR (mm)	1.37 (0.71, 1.85)	1.71 (1.19, 2.86)	0.01*	1.71 (1.53, 2.56)	2.52 (1.76, 3.69)	0.04*	1.57 (0.82, 1.94)	2.07 (1.37, 2.96)	0.00*	
fAT (mm)	3.07 (2.68, 3.49)	3.27 (2.54, 4.16)	0.56	4.22 (3.58, 5.69)	4.18 (2.79, 4.92)	0.72	3.24 (2.84, 4.17)	3.48 (2.60, 4.69)	0.91	
LPL (mm)	7.23 (6.42, 8.93)	7.20 (5.61, 9.26)	0.44	10.33 (7.72, 13.00)	8.52 (5.62, 9.81)	0.07	7.63 (6.64, 10.58)	8.01 (5.62, 9.76)	0.12	
LPR (mm)	6.33 (5.70, 8.31)	7.27 (5.95, 10.03)	0.14	9.51 (6.69, 11.28)	10.34 (7.00, 12.74)	0.42	7.03 (5.76, 9.64)	8.58 (6.24, 11.66)	0.18	
TIL (%)	124% (112%, 134%)	107% (103%, 122%)	0.00*	133% (118%, 154%)	107% (103%, 127%)	0.01*	124% (114%, 144%)	107% (103%, 123%)	0.00*	
TIR (%)	109% (105%, 119%)	122% (107%, 131%)	0.01*	118% ± 0.09	135% ± 0.09	0.00*	113% (106%, 121%)	125% (118%, 136%)	0.00*	
RH (%)	22% (19%, 27%)	23% (18%, 31%)	0.26	40% (30%, 55%)	50% (26%, 56%)	0.70	0.26% (20%, 34%)	28% (20%, 47%)	0.45	
RL/D (%)	52% (47%, 57%)	50% (48%, 55%)	0.43	60% ± 0.19	59% ± 0.16	0.83	52% (47%, 58%)	51% (48%, 57%)	0.57	
Notes.

* Significant difference.

Continuous variables, after tests for normality and homogeneity of variance, are expressed as mean ± SD or median (Q1–Q3).

Multifactorial logistic regression of the total group

Six variables (φ1, φ2, fAL, fAR, TIL, TIR) were integrated into a multifactorial logistic regression analysis, revealing that φ2, fAL, and fAR emerge as independent risk factors for CI formation. A lower φ2 was associated with increased CI risk, with statistical significance (95% CI [0.93–0.99]). Conversely, a higher fAL correlated with elevated CI risk showing statistical significance (95% CI [1.51–4.87]). Similarly, a lower fAR was linked to higher CI risk, exhibiting statistical significance (95% CI [0.24–0.68]).

The Hosmer–Lemeshow test yielded a p-value of 0.43, which is greater than 0.05, indicating a good fit between the established model and actual data. The Omnibus test of model coefficients demonstrated statistical significance for the logistic model (χ2 = 32.60, p < 0.05), enabling correct classification of 77.5% of cases. The regression model equation obtained was ln(P/1-P) = 3.87−0.04* φ2+ fAL −0.9* fAR, where P represents the probability of having CI as 1, and 1-P represents the probability of having CI as 0, as depicted in Table 5.

Table 5 Multivariable logistic regression analysis for the prediction of CI.

Parameter	B	SE	Waldχ2	DF	P	OR	95% CI	
φ2(∘)	−0.04	0.02	5.72	1	0.02	0.96	0.93–0.99	
fAL (mm)	1.00	0.30	11.06	1	<0.01	2.71	1.51–4.87	
fAR (mm)	−0.90	0.26	11.96	1	<0.01	0.41	0.24–0.68	
Constant	3.87	1.59	5.89	1	0.02	47.80	—	

ROC curve analysis

The logistic model, φ2, fAL, and fAR were subjected to ROC curve analysis. As shown in Table 6 and Fig. 5, the logistic model had an AUC of 0.83, sensitivity of 0.71, and specificity of 0.84. The AUC for φ2 was 0.68 with a sensitivity of 0.74 and specificity of 0.65. For fAL, the AUC was 0.72 with a sensitivity of 0.53 and specificity of 0.93. The AUC for fAR was 0.70 with a sensitivity of 0.56 and specificity of 0.76. Compared to individual factors such as φ2, fAL, and fAR, the predictive regression model constructed exhibited superior capability in predicting CI.

Table 6 ROC curve analysis.

Parameter	AUC	95% CI	Sensitivity	Specificity	Optimal cutoff value (Youden index)	
Logistic model	0.83	0.74∼0.91	71%	84%	—	
φ2 (∘)	0.68	0.56∼0.79	74%	65%	92.15	
fAL (mm)	0.72	0.61∼0.84	53%	93%	1.15	
fAR (mm)	0.70	0.59∼0.81	56%	76%	1.94	

Figure 5 ROC curve for predicting cerebral infarction.

Discussion

The cerebral arterial circle, also known as the circle of Willis, is a complex arterial network located at the base of the brain, providing critical blood supply to the cerebral and cerebellar tissues. According to relevant studies, the incidence of CAF is reported to be 1.1%, with the majority occurring in the basilar artery (BA) (Sogawa et al., 2013; Wu et al., 2020). A retrospective analysis of MRA images of 3,327 patients identified 18 intracranial VA (0.54% prevalence), 69 BA (2.07%), and six vertebrobasilar junctions (0.18%), yielding an overall fenestration rate of 2.77% in the intracranial VB system (Uchino et al., 2012). Another study examining cranial MRA images from 891 patients found 11 fenestrations (1.2%) in the A1 and/or A2 segments (Sharma, Verma & Adithan, 2023). Variations in the ACoA have been reported with an incidence of approximately 0.95% (1/105) (Triantafyllou et al., 2024). Fenestration of the MCA is also rare, with prior research identifying two cases (0.5%) among 425 MRA examinations (Uchino et al., 2000).

Among the recruitment cases in the current research, the incidence rate of BA is 67.4%. Simultaneously, BA fenestrations occurred in the proximal segment in 54 cases (90%), in the middle segment in five cases (8.3%), and in the distal segment in one case (1.7%). The occurrence rate of BA fenestrations is highest in the proximal segment. These are well corresponded to this preferential location distribution of arterial fenestration. The link between cerebral infarction (CI) and CAF has long been a topic of speculation and whether fenestration is related to CI remains controversial. One clinical research enrolled 280 cerebrovascular fenestrations concluded that vertebrobasilar fenestrations are more related to cerebrovascular diseases, especially stroke (Ye et al., 2021). It is postulated that this association is predominantly linked to irregularities in the lateral and medial wall structure of fenestration that may alter hemodynamics around fenestration, and hemodynamic abnormalities result in pathological changes in the arterial wall vice versa leading to transient ischemic attacks (Hirai et al., 2021; Omotoso et al., 2022). CI associated with CAF manifest acutely have a short therapeutic window and often result in severe complications that permanently affect the quality of life of patients. Hence, understanding the mechanisms underlying the occurrence of CI accompanied by CAF is of significant clinical importance for timely prediction, diagnosis, and intervention. Arterial morphology regulates flow patterns, thus a thorough understanding of the anatomy and knowledge of anatomical variations of the fenestration is essential for assessing neurologic syndromes and preoperative planning. In this study, we proposed a series of indices to quantify the morphological characteristics of fenestration per se and its connecting arteries, and attempted to identify potential geometrical markers around the fenestration structure for indicating CI risk.

The hemodynamic status around fenestration is still unclear, and few studies focus on the hemodynamic stimuli brought by the fenestration structure that leads to CI. Alteration of flow patterns, e.g., abrupt changes in flow direction, occurs at the initiation and confluence points of fenestrations sites most frequently, transitioning from regular laminar flow to turbulent or erratic flow. Subsequently, components that promote atherosclerosis are transported to the vascular inner wall at these location, resulting in prolonged collision and friction between these components and endothelial cells. The accumulation of harmful substances over time alters the mechanical properties of vessel walls, making fenestration sites and distal vascular segments more prone to lipid deposition and plaque formation, ultimately leading to CI events (Ye et al., 2021). Our results indicate that in the Infarction Group, both φ1 and φ2 were smaller compared to the Control Group. This leads to jet-like flow as buffering angles are reduced at the fenestration site, increasing instability of blood flow along with increased loss of kinetic energy. This, in turn, shortens the proliferation and apoptosis cycle of endothelial cells and enhances their lipid uptake capacity, disrupting their normal physiological processes. The imbalance between protective and injurious factors within the endothelium leads to vascular wall degeneration and increases the likelihood and severity of inflammatory reactions (Tong et al., 2021). One study demonstrated that the flow division from vertebral artery occurs at the fenestration starting site in the CI cases while blood from the vertebral artery directly go through fenestration channel on the corresponding side without flow exchange in control group (Mei et al., 2022). In this research, reduced φ1 and φ2 in the CI group lead to a higher possibility of flow exchange through the fenestration, especially for φ1, which is coincide with our previous hemodynamic study. The frequent changes in blood flow direction the transition to turbulent flow patterns cause heightened friction between blood flow and the vessel wall (Sayed et al., 2024; Szajer & Ho-Shon, 2018). This leads to endothelial cell damage, insufficient recruitment of vasodilators and vasoconstrictors, vascular wall degradation, increased likelihood of inflammatory reactions, and the formation of atherosclerotic plaques in the vessel wall (Tütüncü et al., 2014; Zhang et al., 2014) Studies have shown that vascular regions with high blood flow velocity are more susceptible to the deposition of atherogenic factors like lipoproteins and monocytes, which can accumulate around the vessel wall. In areas of slow blood flow within the vessel wall, these factors move slowly, increasing the contact time with endothelial cells and promoting intimal thickening (Ma et al., 2023).

We discovered that both fAL and TIL were greater in the CI group for CAFs than those in the control group. In contrast, fAR and TIR were smaller than those in the control group. Our results also showed that the number of cases with left dominance of fenestration arterial vessels in the Infarction Group and its subgroups exceeds that of the control group. This suggests that the presence of fenestration deformities, i.e., left-leaning fenestration, leads to unfavourable flow alterations that may contribute to the CI occurrence. Jeong, Kwak & Cho (2008) demonstrated that the atherothrombotic process could be initiated at an arterial branching site and more easily progress to a lateral wall of the smaller limb of the fenestration if the limbs are asymmetric in size. Another research also concluded that more complicated hemorheology is revealed around the fenestration driven by more branched and bends, and the different diameters of fenestration limbs may also increase the incidence of eddies and plaques (Ye et al., 2021). Our results also suggest that asymmetric fenestration limbs are prone to atherothrombotic formation. Therefore, we illustrated diagrams relatedto hemodynamic environment in controls and fenestration at the vertebral-basilar junction (most common) (Fig. 6), to preliminarily demonstrate vessels of varying diameters, uneven blood flow velocities, and turbulence caused by convergence of blood flows from different directions. These factors contribute to the formation of atherosclerotic plaques, potentially leading to the occurrence of CI. Interestingly, our results show that right-side curvature length and tortuosity were smaller in the both Non_BA Infarction Group and BA Infarction Group, indicating that a higher flow exchange occurs at the fenestrated segment and augment the transition from laminar flow to turbulence flow patterns. The basilar artery, formed by the convergence of the bilateral vertebral arteries, is predominantly supplied by the left vertebral artery in most individuals, leading to asymmetric blood flow into the basilar artery, resulting in greater impact on the right wall of the BA (Zhang et al., 2019a; Hong et al., 2009). With the involvement of fenestration, the greater flow rate from the left vertebral artery is forced to be divided into two limbs where the asymmetric structure enhances the unbalanced flow distribution. It is worth noting that this flow predominance of the left vertebral artery may have high correspondence with the left-leaning fenestration type we proposed here, and further hemodynamic research is needed.

Figure 6 Line diagrams showing the complex hemodynamics of the vertebral-basilar arterial system.

Fenestration group in the proximal segment of the basilar artery (A), enlarged model of the fenestration group in the proximal segment of the basilar artery (B), diagram of normal arteries (C), enlarged model of normal arteries (D). Black arrows and blue arrows represent blood flow from the left vertebral artery and right vertebral artery, respectively. Solid lines and dashed lines indicate different blood flow velocities. The red oval area indicates vortices generated by the convergence of blood flow. BA, basilar artery; VA, vertebral artery.

Our study also showed that the number of cases with atherosclerosis in the Infarction Group and its subgroups exceeded that of the corresponding control groups. Asymmetric fenestrated vessels lead to overall blood flow instability, and the flow turbulence motivates endothelial cell dysfunction, disrupts the internal elastic lamina, and recruits a large number of inflammatory cells to vulnerable vascular sites to trigger an inflammatory cascade reaction, inducing medial hypoxia, vascular necrosis, etc. Various inflammatory mediators infiltrate the vessel wall, accompanying significant inflammatory responses, leading to further structural and functional damage to the media, thinning of the vessel wall, a significant reduction in strength, and promoting malignant vascular wall remodelling (Nishikata et al., 2004; Van Rooij et al., 2015). Simultaneously, larger fAL and TIL in the CI group result in a longer remaining path for the small fraction of blood flow, which is prone to stasis and accumulation on the vessel wall. This disrupts collagen fibres and elastic fibres in the vessel wall, impairing the arterial ability to maintain stable contraction and relaxation, further promoting malignant remodelling of the vessel wall, along with plaque formation at the fenestration site and its distal end (Zhang et al., 2022; Wu et al., 2018).

This study, based on CAF MRA images, established an individualized model to acquire morphological parameter indices. It was found that φ2, fL, and fAR were independent risk factors influencing the occurrence of CI, with φ2 having an AUC of 0.68, sensitivity of 0.74, and specificity of 0.65; fAL having an AUC of 0.72, sensitivity of 0.53, and specificity of 0.93; and fAR having an AUC of 0.70, sensitivity of 0.56, and specificity of 0.76.

Furthermore, based on the preliminary establishment of the regression model using morphological parameters, the predictive performance of this model was analyzed by comparing the ROC curves of the regression model with φ2, fAL, and fAR. The research results indicated that, compared to independent risk factors, the AUC of this regression predictive model was 0.83, demonstrating a good predictive performance. Given the multiple complications of CI and the severity of long-term damage in survivors, when this model is applied in clinical settings, it can help identify individuals at risk of CI occurrence at an early stage.

The current study suffers from a limited dataset and a single centre, and the quantity of non-vertebrobasilar cases does not precisely align with the number of vertebrobasilar cases. A more extensive collection of fenestration data from various locations with balanced scales can achieve more unbiased results. Nevertheless, this article carefully separates cases into distinct groups and analyzes the data from BA group and Non_BA group, both groups exhibit consistent morphological features. Moreover, the methodology used in this research primarily focuses on morphological analysis; further studies are needed to delve into the design of hemodynamic analyses, aiming to unravel intricate hemodynamic mechanisms influenced by the morphological features previously examined. Additionally, during the selection stage for vascular model establishment, we excluded cases with severe atherosclerotic stenosis as severe atherosclerosis can lead to poor model establishment and inaccurate generation of vascular centerlines. Patients with moderate or severe atherosclerosis are usually detected and clinically monitored early in the disease course, receiving interventions or conservative treatments. However, there are few studies on fenestrations with mild or insignificant atherosclerosis, and such cases often receive less early attention and clinical consideration. Therefore, our study mainly includes cases with mild atherosclerosis (which result in better modeling outcomes), providing insights into potential etiologies and pathogenic mechanisms. The information provided by TOF MRA regarding plaque characteristics indicative of mild atherosclerosis is limited, and it cannot offer detailed information about intraplaque hemorrhage, among other aspects. Additionally, due to the complexity of vascular supply and compensation in brain tissue, some CI cases were determined to be caused by fenestrated vessels by experienced imaging and clinical physicians through a double-blind assessment. CI has multifactorial origins, with atherosclerosis widely recognized as a major cause. Therefore, our analysis is also based on this assumption, proposing new CI analysis parameters based on the morphological characteristics of CAF, and speculating on the possible relationship between fenestration and CI, expanding on previous studies. We hope these new introducing parameters can provide new perspectives for CI analysis and intervention to aid clinical practice in early intervention. We also plan to include plaque information and other aspects in future studies, such as using 4D Flow imaging technology, to collect comprehensive data on plaque characteristics and continue in-depth research.

Conclusion

This preliminary study indicates a close correlation between the morphological parameters of CAF and the occurrence of cerebral infarction, indicated by a geometrical marker we proposed, i.e., left-leaning fenestration type. It is crucial to quantify the morphological characteristics of fenestration per se and its connecting arteries using specific indices and attempt to identify potential geometric markers around the fenestration structure for indicating CI risk. Furthermore, the regression predictive model established in this study demonstrates a good predictive performance, enabling early prediction of CI occurrence in fenestrated patients and facilitating early diagnosis of CI.

Supplemental Information

Supplemental Information 1 highlights

Supplemental Information 2 Raw data

Additional Information and Declarations

Competing Interests

Author Contributions

Human Ethics

Data Availability

The authors declare there are no competing interests.

Yuqian Mei conceived and designed the experiments, performed the experiments, analyzed the data, prepared figures and/or tables, authored or reviewed drafts of the article, and approved the final draft.

Xiaoqin Chen conceived and designed the experiments, performed the experiments, analyzed the data, prepared figures and/or tables, authored or reviewed drafts of the article, data curation, and approved the final draft.

Yao Zhang conceived and designed the experiments, authored or reviewed drafts of the article, and approved the final draft.

Yanling Wang performed the experiments, authored or reviewed drafts of the article, and approved the final draft.

Bo Wu analyzed the data, prepared figures and/or tables, and approved the final draft.

Mingcheng Hu analyzed the data, authored or reviewed drafts of the article, and approved the final draft.

Quan Bao analyzed the data, authored or reviewed drafts of the article, and approved the final draft.

The following information was supplied relating to ethical approvals (i.e., approving body and any reference numbers):

Mudanjiang Medical University granted ethical approval for this study to be conducted within its facilities (Ethical Application Ref: 202054).

The following information was supplied regarding data availability:

The raw measurements are available in the Supplementary Files.

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
