# Peer review of "Geometrical determinants of cerebral artery fenestration for cerebral infarction"

_PeerJ, doi:10.7717/peerj.18774_

## Round 0.1 · original submission · Minor Revisions

Please revise your manuscript according to the reviewers' comments.

Yours,
Yoshi
Prof. Yoshinori Marunaka, M.D., Ph.D.

·

Basic reporting

To my opinion, the authors presented a very interesting and detailed investigation of the cerebral arterial circle fenestration variants. They have used professional English throughout their manuscript. Everything was submitted properly along with their manuscript, while the figures are of high quality.
However, I want to point out some minor basic issues. Although the authors present results of the cerebral fenestrations, they are only depicting basilar artery figures. Therefore, I recommend to enhance your manuscript with figures from the other locations, as they are mentioning in Table 1.
The references that they are provided are up-to-date; however, a similar comment with the previous one arose. To my opinion, as an anatomists, the authors did not report the prevalence of cerebral arterial circle fenestrations which is of importance when the main point of your manuscript is these variations (I will provide the most adequate references in the following sections). Thus, I recommend to add a relevant paragraph or section to your discussion.

Experimental design

The primary research question, methods and results were well-defined by the authors.

Validity of the findings

The results and their conclusions are interesting, well described and they are adding something new to the current literature.

Additional comments

References Suggestions:
- VA + BA fenestration: Uchino A, et al. Fenestrations of the intracranial vertebrobasilar system diagnosed by MR angiography. Neuroradiology. 2012 May;54(5):445-50. doi: 10.1007/s00234-011-0903-x.
- ACA fenestration: Uchino A, et al. Anterior cerebral artery variations detected by MR angiography. Neuroradiology. 2006 Sep;48(9):647-52. doi: 10.1007/s00234-006-0110-3.
- AComA fenestration: Triantafyllou G, et al. The anterior communicating artery variants: a meta-analysis with a proposed classification system. Surg Radiol Anat. 2024 May;46(5):697-716. doi: 10.1007/s00276-024-03336-7.
- MCA fenestration: Uchino A, et al. Middle cerebral artery variations detected by magnetic resonance angiography. Eur Radiol. 2000;10(4):560-3. doi: 10.1007/s003300050960.

dReviewer 2 ·

Basic reporting

no comment

Experimental design

no comment

Validity of the findings

no comment

Additional comments

It is a good study that will fill a missing area in the literature. It would be better if the minor revisions I sent in the attachment are corrected.

Annotated reviews are not available for download in order to protect the identity of reviewers who chose to remain anonymous.

---

## Round 0.2 · accepted · Accept

Congratulations!
Yours,
Yoshi
Prof. Yoshinori Marunaka, M.D., Ph.D.

·

Basic reporting

The authors addressed all reported issues.
Congratulations!

Experimental design

The authors addressed all reported issues.
Congratulations!

Validity of the findings

The authors addressed all reported issues.
Congratulations!

Reviewer 2 ·

Basic reporting

No comment

Experimental design

No comment

Validity of the findings

No comment

Additional comments

Dear Editor,
The authors have made the article more understandable by taking the suggested corrections into consideration. If you find it appropriate, it can be accepted for publication as is.